# Task Shifting and Task Sharing Implementation in Africa: A Scoping Review on Rationale and Scope

**DOI:** 10.3390/healthcare11081200

**Published:** 2023-04-21

**Authors:** Sunny C. Okoroafor, Christmal Dela Christmals

**Affiliations:** 1Universal Health Coverage—Life Course Cluster, World Health Organization Country Office for Uganda, Kampala, Uganda; 2Centre for Health Professions Education, Faculty of Health Sciences, North-West University, Potchefstroom Campus, Building PC-G16, Office 101, 11 Hoffman Street, Potchefstroom 2520, South Africa

**Keywords:** task shifting, task sharing, rationale, scope, health service delivery, optimal utilization of health workers, access to health services

## Abstract

Numerous studies have reported task shifting and task sharing due to various reasons and with varied scopes of health services, either task-shifted or -shared. However, very few studies have mapped the evidence on task shifting and task sharing. We conducted a scoping review to synthesize evidence on the rationale and scope of task shifting and task sharing in Africa. We identified peer-reviewed papers from PubMed, Scopus, and CINAHL bibliographic databases. Studies that met the eligibility criteria were charted to document data on the rationale for task shifting and task sharing, and the scope of tasks shifted or shared in Africa. The charted data were thematically analyzed. Sixty-one studies met the eligibility criteria, with fifty-three providing insights on the rationale and scope of task shifting and task sharing, and seven on the scope and one on rationale, respectively. The rationales for task shifting and task sharing were health worker shortages, to optimally utilize existing health workers, and to expand access to health services. The scope of health services shifted or shared in 23 countries were HIV/AIDS, tuberculosis, hypertension, diabetes, mental health, eyecare, maternal and child health, sexual and reproductive health, surgical care, medicines’ management, and emergency care. Task shifting and task sharing are widely implemented in Africa across various health services contexts towards ensuring access to health services.

## 1. Introduction

Task shifting and task sharing are being implemented in several countries to efficiently utilize existing health workers to improve the access of the population to quality health services. Access of populations to quality healthcare is essential in achieving Universal Health Coverage (UHC) and the health and health-related Sustainable Development Goals (SDG) [1,2,3]. Task shifting is defined by the World Health Organization (WHO) as “the rational redistribution of tasks among health workforce teams”, from trained and qualified health workers to other health workers with shorter training duration to maximize the available health workforce [2]. In task shifting, tasks are delegated or transferred, and in task sharing, tasks are delivered collaboratively by different staff categories [3]. This approach is implemented in many countries globally, including Africa, where there remains a persistent health workforce shortage and deficient access to healthcare [4,5].

Africa faces numerous health workforce challenges that are contributing to the health indices and systems’ performance of countries in the continent [6]. These challenges are also impacting negatively on the functionality and the resilience of the health system [7,8], and the attainment of key population outcomes [9]. These challenges, which are quite broad and have contextual specificities, include weak health and health workforce leadership, governance and stewardship mechanisms, and management systems, as well as poor regulation, and evidence generation and use mechanisms [6,10]. Furthermore, there is a persistent low stock of qualified and skilled health workers, inequitable distribution of existing ones, marked inequalities in education, employment, and population needs, and poor work environments at various levels of the health system [6,11].

The impact of the aforementioned challenges includes a high deficit in the health workforce in most countries in Africa. The Africa Regional average density of doctors, nurses, and midwives per 1000 population in 2018 was 1.55, which is lower than the SDG index threshold of 4.45 [4]. Coping with the persistent deficit which has been ever-present over the years informed the implementation of formally task shifting and task sharing, and informally in levels of service delivery and programs. Its implementation varies widely, with several countries currently implementing the approach and others planning to commence implementation. For those implementing, in the early stages of implementation or planning to commence implementation, evidence on circumstances that should inform its implementation and the scopes of tasks that could be task shifted or shared is pertinent.

Numerous studies have reported task shifting and task sharing due to various reasons and with varied scopes of health services, either task-shifted or -shared. However, very few studies have holistically mapped the evidence on task shifting and task sharing, with most focusing on specific programs [12,13,14,15]. It is worth reviewing the evidence of task shifting and task sharing for integrated health service delivery, focusing on the rationale and scope of tasks. Therefore, this scoping review aimed to synthesize evidence on the rationale and scope of task shifting and task sharing in Africa.

## 2. Materials and Methods

This scoping review was conducted using the enhanced Arksey and O’Malley’s framework for scoping reviews [16,17]. The review questions were: (1) What are the documented rationales for task shifting and task sharing? (2) What scope of tasks were shifted or shared to improve the access of the population to health services in Africa? To answer these questions, the Joanna Briggs Institute’s population, concept, and context (PCC) framework presented in Table 1 was adopted.

We identified relevant studies by applying the search strategy in Table 2 to obtain peer-reviewed papers from the PubMed, Scopus, and CINAHL bibliographic databases. We considered quantitative, qualitative, and mixed-methods studies, as well as review and perspective papers on task shifting or sharing for integrated health service delivery. We also considered articles published from 2010 to 2021 to obtain a wide range of contemporary information.

All articles obtained from the literature search, which was conducted independently by the authors, were downloaded into the Mendeley reference manager, with duplicates removed. The title and abstract were screened independently by the authors, with discrepancies discussed and a consensus reached. Afterwards, the full text was reviewed based on the eligibility criteria for inclusion. The inclusion criteria were: (1) quantitative, qualitative, and mixed-methods studies on task shifting or sharing for integrated health service delivery in Africa, (2) review and perspective papers on task shifting or sharing for integrated health service delivery in Africa, (3) full-text articles are written in the English language and are accessible, and (4) articles were published from 2010 to 2021. The exclusion criteria were: (1) papers written in other languages, (2) full texts were not accessible, and (3) papers were news articles, editorials, commentaries, and letters to the editor that did not involve primary data and did not provide insights into the rationale, scope, and health professions education for task shifting and task sharing in Africa.

The first author (S.C.O.) used a data extraction matrix to extract information on the author and year of publication, country name, study design, study setting, and population. Additionally extracted were the key findings on the rationale and scope of task shifting and task sharing, health service context, and the level of care. As had been carried out in other studies, we extracted text verbatim on themes, concepts, and categories relevant to the research questions [18,19] into the extraction matrix for analysis using the thematic analysis approach. The second author randomly reviewed the charting for 50% of the studies, with an agreement reached on the charting output.

We analyzed the characteristics of the papers based on the extracts using descriptive statistics. We analyzed the qualitative content of the papers using thematic analysis [20,21], with synthesized information described narratively based on information on the rationale and scope of task shifting and task sharing in Africa. We reported the findings using the Preferred Reporting Items for Systematic Reviews and Meta-Analyses guidelines Extension for Scoping Review reporting standards (Table A1).

## 3. Results

### 3.1. Description of Studies

We identified 926 records from the databases CINAHL (*n* = 197), PubMed (*n* = 567), and Scopus (n = 162). After duplicates were removed, 510 records were screened by titles and abstracts, from which 141 articles’ full text were assessed for eligibility (Figure 1). Sixty-one original research papers were included in the final review, with nine (15%) mixed-methods studies, forty-two (69%) quantitative, and ten (16%) qualitative (Table 2). In all, 80 papers were excluded following the full-text review, with 29 being books, editorials, commentaries, and letters to the editor without primary data, 47 papers providing no insights on the rationale and scope of task shifting and task sharing, and 4 not being conducted in Africa.

### 3.2. Characteristics of the Included Studies

The main characteristics of the included papers are presented in Table 3, with details in Table 4. The highest proportion of the papers was published in 2017 (18%), with the lowest proportion published in 2019 (3%). Of the included papers, 2 were multi-country, and 22 were conducted in different countries in Africa. Thirteen percent (*n* = 8) were conducted in Uganda, and ten percent (*n* = 6) were conducted in Ghana and Kenya. Of the reviewed papers, 87% (*n* = 53) provided insights on the rationale and scope of task shifting and task sharing, with 11% (*n* = 7) providing insights on scope only, and 2% (*n* = 1) on rationale only. Furthermore, 59% (*n* = 36) of the studies reported task shifting and task sharing practices at the community and primary level of care, with 3% (*n* = 2) of studies reporting practices at the tertiary level of care.

### 3.3. Main Themes from the Included Studies

The main theme and subthemes are presented in Table 5 and described in subsequent sections.

#### 3.3.1. Rationale for Task Shifting and Task Sharing

Fifty-four studies [22,23,24,25,26,27,28,29,30,31,32,33,34,35,36,37,38,39,40,41,42,43,44,45,46,47,48,49,50,51,52,53,54,55,56,57,58,59,60,61,62,63,64,65,66,67,68,69,70,71,72,73,74] included under this theme provided insights on the rationale for the implementation of task shifting and task sharing in Africa, as presented in Table 6.

##### Health Worker Shortages

Forty studies [22,23,24,25,26,27,28,29,30,31,32,33,34,35,36,37] reported the rationale for task shifting and task sharing implementation to be due to a shortage of health workers. The cadres reported to be inadequate to inform task shifting and task sharing in the various countries included physicians [38,39,40,41,42,43], nurses [44,45,46], midwives [46], surgical specialists [47,48], eye care specialists [75,76], pharmaceutical staff [49], emergency care practitioners [27], psychiatrists [50,51,52,53], psychologists [51], and pathologists [54]. Insights on the reported shortages were provided in four studies. A study in Malawi [22] stated that the shortage was prominent by the level of care and geographical location. The studies in Cameroon [41], Ethiopia [55], and Nigeria [56] reported that the shortage of physicians and nurses was more prominent in rural areas. Two other Nigerian studies reported shortages being marked at the primary level of care [45,77]. Further insight on the impact on health services’ delivery was provided in two Ugandan studies [57,58]. These studies highlighted that the health worker shortage meant the high demand for health services could not be adequately met.

##### Optimally Utilize Existing Health Workers

Seven [22,59,60,61,62,63,64] studies reported the rationale for task shifting and sharing implementation to include the need to optimally utilize the available human resources within a health service level to deliver health services. The study in Benin [59] implemented task shifting to expand the role of existing lay nurse aides to conduct counselling in maternal and newborn care. In a study conducted in Kenya [60], task shifting was implemented in HIV/AIDS care to optimally utilize existing health workers to improve outcomes without increasing resources. A Nigerian study [61] reported the implementation of task shifting to reduce the waiting time for accessing services from doctors by expanding the role of existing nurses. In Zambia [62], the rationale for implementing task shifting and sharing was to utilize existing peer educators in expanding the delivery of health services. A study in Tanzania [63] reported a task shifting implementation in preventing mother-to-child transmission (PMTCT) service delivery to optimally utilize existing health workers by reducing nurses’ workload and health system costs.

##### Expand Access to Health Services

Twelve [22] studies reported the rationale for task shifting and sharing implementation to be to ensure increased access to health services at other levels of care and geographical locations, specifically, rural areas. The countries where an expansion of health services to other levels of care was reported include Ghana, Kenya, Madagascar, Malawi, Namibia, Swaziland, and Uganda. A study in Ghana [68] reported the implementation of task sharing to expand the service delivery of hypertension management and control to lower levels of care. In Kenya [45], the rationale for implementing task shifting was to increase access to non-communicable diseases (NCD) services (hypertension, diabetes mellitus type 2, epilepsy, asthma, and sickle cell) in primary healthcare settings. Another study in Kenya [74] reported implementing task shifting to improve access to mental health interventions at the community level. The study in Madagascar [66] reported that task shifting of the provision of contraceptives was implemented to expand access of community members through community health workers. A study in Malawi [67] reported task shifting of primary mental healthcare to the community level to improve the access of community members. In a Namibian study [73], task shifting was implemented to scale-up point-of-care CD4+ testing in HIV counselling and testing settings in public health facilities. A study in Swaziland [72] reported the implementation of task sharing to decentralize antiretroviral (ART) provision to improve access. A Ugandan study [71] reported the implementation of task shifting for tuberculosis to improve the access of community members to directly observed treatment short course (DOTS).

Studies from Cameroon, Eswatini, Ethiopia, and Senegal reported the implementation of task shifting and task sharing to expand health services to rural areas. In a study conducted in Cameroon [65], task shifting was implemented to increase the access of the rural population to adequate hypertension and diabetes care. A study in Eswatini [35] reported the implementation of task shifting tuberculosis management to improve access in rural areas. A study in Ethiopia [69] reported the task sharing implementation to improve access to family planning in rural areas. In a study in Senegal [70], task sharing to increase the use of long-acting reversible contraceptives was performed to improve the access to family planning in rural areas.

#### 3.3.2. Scope of Shifted and Shared Tasks

Sixty reviewed studies reported a range of tasks that were either task-shifted or task-shared in various health services contexts in Africa (Table 6).

##### HIV/AIDS Care

Twenty-three studies reported task shifting and task sharing of HIV/AIDS services, as shown in Table 6. The HIV/AIDS service areas reported included HIV counselling and testing, ART, PMTCT, and care and support. Task shifting and sharing of HIV counselling and testing were reported in studies in Botswana [24], Malawi [22], Namibia [73], Swaziland [72], and Zambia [26]. The beneficiary cadres included laypersons [24], health surveillance assistant (HSA)/lay counsellor [22,26,72], nurses [26,73], and laboratory personnel [26]. ART-related task shifting and sharing were reported in Cameroon [41], Ethiopia [78], Kenya [60], and Malawi [22,44], with beneficiary cadres including nurses [22,41,60,78], health officers [78], clinical officers [22,60], medical assistants [22], and health surveillance assistants (HSAs) [44]. The task shifting and task sharing of care and support services by community health workers [79], people living with HIV/AIDS (PLWAs) as community care coordinators [80], laypersons [29], maternal and child nurses [64], and peer educators [62] was reported in Kenya [79,80], Malawi [29], Mozambique [64], and Zambia [62].

##### Hypertension Management

Five studies reported task shifting and task sharing in the management of hypertension. A study conducted in Cameroon [65] reported a task shifting of integrated management of hypertension to non-physician clinicians (nurses) in a rural setting to improve the access to care for hypertension. In Kenya [45], NCD service delivery, including hypertension, was task-shifted to nurses in primary healthcare settings. In a study conducted in the Democratic Republic of Congo (DRC) [31], hypertension management was task-shifted to nurses. In a Ghanian study [68], hypertension management was task-shifted to community health nurses (CHNs) and enrolled nurses (ENs). The study conducted in Mozambique [37] reported that initial screening and initiation of obstetric emergency care for pre-eclampsia/eclampsia was task-shifted to community health workers.

##### Diabetes Management

Two studies reported task shifting and task sharing in diabetes management. The first study from Cameroon [65] reported a task shifting of integrated management of diabetes to nurses in a rural setting to improve access to type 2 diabetes care. The second study, that was conducted in Kenya [45], reported task shifting of NCD services, including for diabetes mellitus, to nurses in primary healthcare settings.

##### Mental Health

Ten studies conducted in eight countries reported the scope of shifted or shared tasks to be within the delivery of the mental health services domain. Three studies in Ghana [50,52,53] reported the task shifting of mental healthcare to community mental health workers—community psychiatric nurses (CPNs), clinical psychiatric officers, (CPOs), and community mental health officers (CMHOs). A study in Malawi [67] reported task shifting of primary mental healthcare at the community level to village-based health workers. In a study conducted in Zimbabwe [81], delivery of depression and other common mental disorders (CMD) services, specifically screening and monitoring of CMD and delivery of interventions, were task-shifted to lay workers. The study in Uganda [57] reported task shifting and sharing practices to include psychiatric clinical officers covering the same scope as the psychiatrists. A study in Mozambique [51] reported task shifting of the delivery of psychiatric care to psychiatric technicians due to low numbers of psychiatrists and psychologists. A study in Kenya [74] reported implementing task shifting of community-based family therapy mental health interventions to lay counsellors. A study in South Africa [36] reported task shifting of psychological treatment for perinatal depression to non-specialist community health workers. Another study in South Africa [82] reported task sharing of mental health counselling to non-specialist facility-based counsellors (FBCs).

##### Maternal and Child Healthcare

In six studies from four countries, maternal and child health services were either task-shifted or task-shared. The study in Benin [59] implemented task shifting to expand the role of existing lay nurse aides to conduct counselling in maternal and newborn care. A study in Ethiopia [55] reported the task shifting of comprehensive emergency obstetric care (CEmOC) to non-physician clinicians (NPCs). In another study in Ethiopia [43], task shifting of the conduct of the caesarian section to non-physician surgeons (NPS) was reported. In a study in Nigeria [77], detection of early signs of pre-eclampsia was reported to be task-shifted, albeit informally, to community health extension workers. The study in Uganda [57] reported task shifting and task sharing practices to include midwives conducting manual vacuum extraction, manual removal of the placenta, and manual vacuum aspiration due to a shortage of doctors, and community health workers (CHWs) and community members were involved in the delivery of an expanded program on immunization (EPI) services. Another study in Uganda [39] highlighted that due to the absence of physicians in certain locations, midwives were often the main providers of post-abortion care services.

##### Sexual and Reproductive Health Services

Eight reviewed studies from seven countries reported the task shifting or task sharing of sexual and reproductive health services. A study in Burkina Faso [33] reported task shifting of the provision of oral and injectable contraceptives to community health workers, and implants and intrauterine devices to auxiliary midwives and nurses. A study in Ethiopia [69] reported the implementation of task sharing of the provision of long-acting contraceptive (Implanon) family planning services to health extension workers. A study in Ghana [34] reported the implementation of task sharing of the provision of an intrauterine contraceptive device to community health nurses. The study in Madagascar [66] task-shifted the provision of injectable contraceptives and counselling of patients to community health workers. In a study in Malawi [46], the task of long-acting reversible contraception (LARC) insertion was shifted to community midwife assistants (CMAs). In a study in Nigeria [28], the provision of contraceptive implants was task-shifted to community health extension workers. Another Nigerian study [56] reported the task shifting of the screening for cervical cancer using visual inspection with acetic acid to community health officers and community health extension workers. In a study in Senegal [70], task sharing of long-acting reversible contraceptives (LARC), specifically implants and intrauterine devices, was implemented for nurses, non-clinical family planning counsellors, and community health workers.

##### Eye Care

Task shifting of eye care services was reported in three studies conducted in six countries. A study in Uganda [57] reported task shifting and sharing practices to include ophthalmic clinical officers conducting cataract surgery. A study [75] in Madagascar, Malawi, and Rwanda reported task shifting of primary eye care service delivery to general primary healthcare (PHC) workers (ophthalmic clinical officers). A multi-country study [76] in Kenya, Malawi, and Tanzania also reported task shifting of cataract surgery to non-physician cataract surgeons.

##### Tuberculosis Care

The task shifting of tuberculosis-related care was reported in three studies. A Ugandan study reported the task shifting of directly observed treatment short course (DOTS) to laypersons [71] and nurses [25]. A study in South Africa [42] reported the task sharing of multidrug-resistant tuberculosis (MDR-TB) treatment between clinical nurse practitioners (CNPs) and medical officers (MO). A study in Eswatini [35] reported the task shifting of directly observed treatment (DOT) supervision and administration of intramuscular MDR-TB injections to lay health workers (LHWs)/community treatment supporters (CTS) to improve the access in rural areas.

##### Surgical Care and Procedures

For surgical care and procedures, three studies provided insights into task shifting and sharing practices. A study in Uganda [47] that explored surgical task shifting practices from surgical specialists to non-specialist physicians reported that the practice was already in place. A study in Sierra Leone [48] reported the implementation of task sharing of surgical and obstetric emergencies to associate clinicians/community health officers (CHOs) and junior doctors. In a Kenyan study [54], the conducting of fine-needle aspiration biopsy cytology, bone marrow aspiration, and trephine biopsy was shared by pathologists, medical officers (MO), clinical officers (CO), and technologists.

##### Medicines’ Management

One study from Tanzania [49] reported task shifting of pharmaceutical management to nurses and medical attendants. This was reported to be because of the shortage of pharmaceutical personnel in the country, which impacts negatively on service provision, especially in rural areas.

##### Emergency Care

One study reported task shifting of emergency care-related services. The study from Uganda [27] reported task shifting of acute care for emergency services in a rural setting to nurses. This was pertinent as 84% of the country’s population reside in rural areas and they require access to emergency care.

## 4. Discussion

This scoping review synthesized evidence on the rationale for task shifting and task sharing in Africa. The reported rationales in the reviewed papers were health worker shortages, the need to optimally utilize existing health workers, and the need to expand access to health services.

The health worker shortages were reported to be prominent by the level of care and geographical location, especially in rural areas. The finding on the shortage of health workers matches the reported global shortage of health workers, which is projected at 10 million by 2030 [83] and is more prominent in Africa [84] and rural and remote areas [85]. The implementation of task shifting and task sharing to cope with this is consistent with the literature. The suboptimal funding of public health [86], including service delivery and its impact on the quality of health infrastructure, the availability of health workers, and service delivery, is also reported in the literature. Similarly, the weak investment in HRH development has also been widely reported in Africa, and this is suggested to substantially contribute to the shortage of health workers [87]. Contributing to this is the dearth of attraction and retention strategies in countries, which is also compounding the shortages and resulting in poorly motivated health workers that are also inequitably distributed [85].

We also found that the need to optimally utilize existing health workers was another reported rationale for task shifting and task sharing. An important perspective gained in the study from Kenya [60] was task shifting being implemented to improve populations’ access to health services without increasing resource needs. This is recommended in instances where evidence exists of varied workloads and availability of certain cadres of health workers with low workloads that can take up additional tasks and relieve those with high workloads [88]. This should, however, be informed by an assessment of the workloads, use of the evidence to explore task shifting and task sharing possibilities [89], and adequate capacity building and provision of relevant incentives to beneficiary cadres, in order to ensure quality service delivery.

Another reported rationale for task shifting and task sharing was to expand access to health services to other levels of care and geographical locations, specifically, rural areas. The reported expansion of services to other levels of care and rural areas was implemented in scenarios where the primary cadres responsible for the service delivery were not typically available at the target level of care, based on the service delivery organization model. In the reviewed studies, the reported instances were expanding service delivery to primary [45] and community level settings [66,67,74] and rural areas [35,65,69], for improved access to NCD services, family planning services, and HIV/AIDS and tuberculosis services.

This scoping review also synthesized evidence on the scope of task shifting and task sharing implementation in Africa. Our findings indicate that the health services shifted or shared in 23 countries where reviewed papers emanated include communicable diseases (HIV/AIDS and tuberculosis), NCD (hypertension, diabetes, mental health, and eyecare), maternal and child health, sexual and reproductive health, surgical care, medicines’ management, and emergency care. In addition to these conditions and health services being those contributing to the burden of morbidity and mortality in Africa [90,91], some of these areas, especially HIV/AIDS, tuberculosis, maternal and child health, sexual and reproductive health, and medicines’ management, have received substantive investments from donors and partners in recent years. Perhaps, this investment promoted the task shifting and task sharing practices that were published in the reviewed studies across all levels of the health system, considering most of the studies were funded by projects.

### 4.1. Implications for Policy and Practice

The health system in Africa will likely continue to be faced with a shortage of health workers until urgent steps are taken. The aim should be to urgently strengthen the health system to become adequately resilient [92] and ensure universal access to qualified, skilled, and motivated health workers that are equitably distributed [93]. Until this is achieved, the persistent health worker shortages call for policies aimed at ensuring that adequate numbers of qualified, skilled, and motivated health workers are available to deliver quality health services at all levels of the system in Africa, based on population health needs. Achieving this also requires apposite contextual policies and interventions to attract and retain health workers, with emphasis given to rural and remote areas.

Task shifting and task sharing implementation to optimally utilize existing health workers or expand service delivery using cadres whose primary function differs should be implemented based on contextual needs. Furthermore, evidence from a needs assessment on scope and competencies should be documented [27] and used to inform the capacity building and the provision of necessary job aids [45,72]. These are pertinent in ensuring quality service delivery.

### 4.2. Limitations

The findings should not be considered to provide a complete view of the rationale and scope for task shifting and task sharing in Africa, as this review was based on peer-reviewed literature that met a set of criteria. There is a possibility that there are ongoing practices that either are not published in the peer-reviewed literature or did not meet the inclusion criteria for this review. Additionally, in alignment with the scoping review methodology, we did not critically appraise the evidence provided in this paper. We may have also missed out on task shifting and task sharing practices not published in English. Lastly, although we rigorously searched three databases, we may have missed out on studies that are available in other databases.

## 5. Conclusions

Task shifting and task sharing are widely implemented in Africa across various health services contexts, aimed towards ensuring access to health services. To guide its implementation, populations’ needs should be used to inform the capacity building of beneficiary cadres to ensure that they have the required knowledge, skills, competence, and job aids to guarantee quality service delivery.

## Figures and Tables

**Figure 1 healthcare-11-01200-f001:**
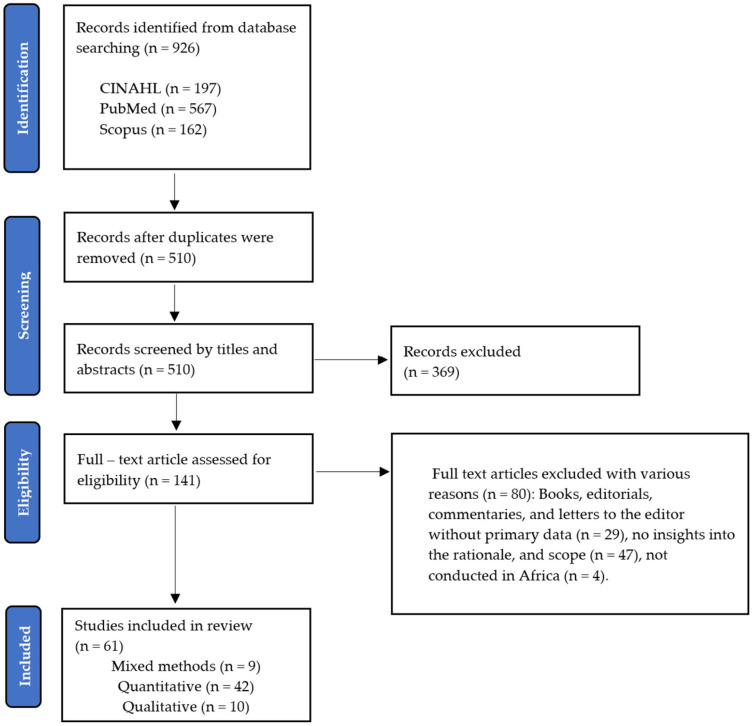
PRISMA-ScR flow diagram.

**Table 1 healthcare-11-01200-t001:** Population, concept, and context (PCC) framework for the scoping review.

Criteria	Component(s)	Explanation
Population (P)	Population Health workforce	Everyone accessing health services. Healthcare workers such as physicians, nurses, midwives, and community health workers that are working as frontline contact in the healthcare system.
Concept (C)	Rationale Scope	Usual reasons or the logical explanations for task shifting and task sharing. Extent or range of services that are task-shifted or -shared amongst various categories of health workers.
Context (C)	Healthcare services African countries	Healthcare services within essential service packages that are either sought or received by the population at any healthcare service delivery level in both public and private sectors. Any country within the African continent.

**Table 2 healthcare-11-01200-t002:** Search strategy.

Source of Literature	Task Shifting/Sharing Terms	Health System/Services Terms	Africa Terms
PubMed	“task shifting” [Title/Abstract] OR “task sharing” [Title/Abstract] AND (“Africa”) [Mesh]	“health system*” [tiab] OR “healthcare system*” [tiab] OR “health care system*” [tiab] OR “healthcare sector*” [tiab] OR “healthcare industr*” [tiab] OR “health industr*” [tiab] OR “health facilit*” [tiab] OR “hospital*” [tiab] OR “healthcare” [tiab] OR “health care” [tiab] OR “health service*” [tiab] OR “healthcare service*” [tiab] OR “health cent*” [tiab] OR “care, health” [tiab] OR “system, health care” [tiab] OR “systems, health care” [tiab] OR “system, healthcare” [tiab]	Africa*[tiab] OR Algeria*[tiab] OR Angola*[tiab] OR Benin*[tiab] OR Botswana*[tiab] OR Burkina Faso [tiab] OR Burundi*[tiab] OR Cape Verde*[tiab] OR Cabo Verde [tiab] OR Cameron*[tiab] OR Cameroon*[tiab] OR Chad*[tiab] OR Comoros*[tiab] OR Congo*[tiab] OR Cote d’Ivoire[tiab] OR Ivory coast [tiab] OR Djibouti*[tiab] OR Egypt*[tiab] OR Eritrea*[tiab] OR Ethiopia*[tiab] OR Gabon*[tiab] OR Gambia*[tiab] OR Ghana*[tiab] OR Guinea*[tiab] OR Kenya*[tiab] OR Lesotho*[tiab] OR Liberia*[tiab] OR Libya*[tiab] OR Madagascar*[tiab] OR Malawi*[tiab] OR Mali*[tiab] OR Maurit*[tiab] OR Morocc*[tiab] OR Mozambiqu*[tiab] OR Namibia*[tiab] OR Niger*[tiab] OR Rwanda*[tiab] OR Senegal*[tiab] OR Seychelles[tiab] OR Sierra Leone*[tiab] OR Somalia*[tiab] OR South Africa*[tiab] OR Sudan*[tiab] OR Swaziland*[tiab] OR Tanzania*[tiab] OR Togo*[tiab] OR Tunisia*[tiab] OR Uganda*[tiab] OR Zambia*[tiab] OR Zimbabwe*[tiab]
CINHAL	TI “task shifting” OR AB “task shifting” OR TI “task sharing” OR AB “task sharing” OR MH “task shifting” OR MH “task sharing”	TI” health system*” OR AB “health system*” OR TI” healthcare system*” OR AB” healthcare system*” OR TI” health care system*” OR AB” health care system*” OR TI” healthcare sector*” OR AB” healthcare sector*” OR TI” health facilit*” OR AB” health facilit*” OR TI “hospital*” OR AB “hospital*” OR TI” healthcare” OR AB” healthcare” OR TI “health care” OR AB “health care” OR TI” health service*” OR AB” health service*” OR TI” healthcare service*” OR AB” healthcare service*” OR TI” health cent*” OR AB” health cent*” OR TI” care, health” OR AB” care, health” OR TI “system, health care” OR AB “system, health care” OR TI “systems, health care” OR AB “systems, health care” OR TI “system, healthcare” OR TI “system, healthcare”	MH Africa OR TI Africa* OR AB Africa* OR TI Algeria* OR AB Algeria* OR TI Angola*OR AB Angola* OR TI Benin*OR AB Benin* OR TI Botswana* OR AB Botswana OR TI “Burkina Faso*” OR AB “Burkina Faso*” OR TI Burundi* OR AB Burundi* OR TI “Cape Verde*” OR AB “Cape Verde*” OR TI Cameron* OR AB Cameron*OR TI Cameroon*OR AB Cameroon* OR TI Chad* OR AB chad* OR TI Comoros* OR AB Comoros* OR TI Congo* OR AB Congo* OR TI “Cote d’Ivoire” OR AB “Cote d’Ivoire” OR TI “Ivory coast” OR AB “Ivory coast” OR TI Djibouti* OR AB Djibouti* OR TI Egypt* OR AB Egypt* OR TI Eritrea* OR AB Eritrea* OR TI Ethiopia* OR AB Ethiopia* OR TI Gabon* OR AB Gabon* OR TI Gambia*OR AB Gambia* OR TI Ghana* OR AB Ghana OR TI Guinea* OR AB Guinea* OR TI Kenya* OR AB Kenya* OR TI Lesotho* OR Lesotho* OR TI Liberia* OR AB Liberia OR TI Libya* AB Libya* OR TI Madagascar* OR AB Madagascar* OR TI Malawi* OR AB Malawi* OR TI Mali* OR AB Mali* OR TI Maurit* OR AB Maurit* OR TI Morocc* OR AB Morocc* OR TI Mozambiqu* OR AB Mozambiqu* OR TI Namibia* OR AB Namibia* OR TI Niger* OR AB Niger* OR TI Rwanda* OR AB Rwanda* OR TI Senegal* OR AB Senegal* OR TI Seychelles* OR TI Seychelles* OR TI “Sierra Leone*” OR AB “Sierra Leone*” OR TI Somalia* OR AB Somalia” OR TI South Africa* OR AB South Africa* OR TI Sudan* OR AB Sudan* OR TI Swaziland* OR AB Swaziland* OR TI Tanzania* OR AB Tanzania* OR TI Togo* OR AB Togo* OR TI Tunisia* OR AB Tunisia OR TI Uganda* OR AB Uganda* OR TI Zambia* OR TI Zambia* OR TI Zimbabwe* OR AB Zimbabwe*
Scopus	“task shifting” OR “task sharing”	“health system*” OR “healthcare system*” [tiab] OR “health care system*” OR “healthcare sector*” OR “healthcare industr*” OR “health industr*” OR “health facilit*” OR “hospital*” OR “healthcare” OR “health care” OR “health service*” OR “healthcare service*” OR “health cent*” OR “care, health” OR “system, health care” OR “systems, health care” OR “system, healthcare”	TITLE-ABS-KEY (“Africa” OR “Algeria*” OR “Angola*” OR “Benin*” OR “Botswana*” OR “Burkina Faso” OR “Burundi*” OR “Cape Verde*” OR “Cabo Verde” OR “Cameron*” OR “Cameroon*” OR “Chad*” OR “Comoros*” OR “Congo*” OR “Cote d’Ivoire” OR “Ivory coast” OR “Djibou*” OR “Egypt*” OR “Eritrea*” OR “Ethiopia*” OR “Gabon*” OR “Gambia*” OR “Ghana*” OR “Guinea*” OR “Kenya*” OR “Lesotho*” OR “Liberia*” OR “Libya*” OR “Madagascar*” OR “Malawi*” OR “Mali*” OR “Maurit*” OR “Morocc*” OR “Mozambiqu*” OR “Namibia*” OR “Niger*” OR “Rwanda*” OR “Senegal*” OR “Seychelles” OR “Sierra Leone*” OR “Somalia*” OR “South Africa*” OR “Sudan*” OR “Swaziland*” OR “Tanzania*” OR “Togo*” OR “Tunisia*” OR “Uganda*” OR “Zambia*” OR “Zimbabwe*”)

**Table 3 healthcare-11-01200-t003:** Main characteristics of the included papers.

Paper Characteristics	Categories	Results	
		*n*	%
Year of publication	2010	4	7
	2011	5	8
	2012	5	8
	2013	4	7
	2014	6	10
	2015	6	10
	2016	7	11
	2017	11	18
	2018	4	7
	2019	2	3
	2020	4	7
	2021	3	5
Country of study	Benin	1	2
	Botswana	1	2
	Burkina Faso	1	2
	Cameroon	3	5
	Democratic Republic of the Congo	1	2
	Eswatini	1	2
	Ethiopia	4	7
	Ghana	6	10
	Kenya	6	10
	Madagascar	1	2
	Malawi	5	8
	Mozambique	3	5
	Namibia	2	3
	Nigeria	4	7
	Senegal	1	2
	Sierra Leone	1	2
	South Africa	4	7
	Swaziland	1	2
	Tanzania	2	3
	Uganda	8	13
	Zambia	2	3
	Zimbabwe	1	2
	Multi-country	2	3
Type of article	Original research	61	100
Study design	Quantitative research	42	69
	Qualitative research	10	16
	Mixed methods	9	15
Main focus	Rationale	1	2
	Scope	7	11
	Both	53	87
Level of care	Community care level	8	13
	Primary care level	13	21
	Primary and community care level	15	25
	Secondary and primary care level	7	11
	Secondary care level	13	21
	Secondary, primary, and community care level	2	3
	Tertiary care level	2	3
	All levels	1	2

**Table 4 healthcare-11-01200-t004:** Details on study characteristics.

First Author (Year)	Country	Study Title	Study Design and Population	Study Focus	Level of Care
Bemelmans 2010	Malawi	Providing universal access to antiretroviral therapy in Thyolo, Malawi through task shifting and decentralization of HIV⁄AIDS care	Design Cross-sectional study (descriptive) Population: People living with HIV/AIDS Health workers: Nurses to health surveillance assistant (HSA)/lay counsellor, medical doctors, clinical officer, medical assistant, and nurses.	Rationale, Scope	Secondary and primary care level
Kosgei 2010	Kenya	Task shifting in HIV clinics, Western Kenya	Design Cross-sectional study (descriptive)	Rationale, Scope	Primary care level
			Population: People living with HIV/AIDS		
			Health workers: Nurses and clinical officer.		
Labhardt 2010	Cameroon	Task shifting to non-physician clinicians for integrated management of hypertension and diabetes in rural Cameroon: a programme assessment at two years	Design: Cross-sectional study (implementation)	Rationale, Scope	Primary care level
			Population: People requiring hypertension and diabetes care		
			Health workers: Non-physician clinician (NPC)—nurses		
Selke 2010	Kenya	Task-Shifting of antiretroviral delivery from health care workers to persons living with HIV/AIDS: clinical outcomes of a community-based program in Kenya	Design: Prospective cluster-randomized controlled clinical trial Population: People living with HIV/AIDS Health workers: People living with HIV/AIDS (PLWAs) as community care coordinators (CCCs)	Scope	Primary care level
De Wet 2011	South Africa	Exploring task-shifting practices in antiretroviral treatment facilities in the Free State Province, South Africa	Design: Prospective cluster-randomized controlled clinical trial Population: People living with HIV/AIDS Health workers: Nurses and community health workers—‘lay workers’, ‘community care workers’, ‘home-based carers’, ‘directly observed treatment (DOT) supporters’, or ‘lay counselors.	Rationale, Scope	Primary care level
Chibanda 2011	Zimbabwe	Problem-solving therapy for depression and common mental disorders in Zimbabwe: piloting a task-shifting primary mental health care intervention in a population with a high prevalence of people living with HIV	Design: Cross-sectional study (implementation)	Scope	Primary care level
			Population: General population with a high prevalence of people living with HIV		
			Health workers: Lay workers (health promoters)		
Gessessew 2011	Ethiopia	Task shifting and sharing in Tigray, Ethiopia, to achieve comprehensive emergency obstetric care	Design: Cross-sectional study (retrospective review of hospital records)	Scope	Secondary and primary care level
			Population: Women of reproductive age receiving obstetric care		
			Health workers: Non-physician clinicians (NPCs)		
Jennings 2011	Benin	Task shifting in maternal and newborn care: a non-inferiority study examining delegation of antenatal counseling to lay nurse aides supported by job aids in Benin	Design: Non-inferiority quasi-experimental design	Rationale, Scope	Primary care level
			Population: Women of reproductive age receiving maternal and newborn health service		
			Health workers: Nurse-midwives and lay nurse aides		
Umar 2011	Nigeria	Reduction of client waiting time using task shifting in an anti-retroviral clinic at Specialist Hospital Bauchi, Nigeria.	Design: Cross-sectional study (implementation)	Rationale, Scope	Secondary care level
			Population: People living with HIV/AIDS		
			Health workers: Doctors and nurses		
Tweya 2012	Malawi	Task shifting’ in an antiretroviral clinic in Malawi: can health surveillance assistants manage patients safely?	Design: Cross-sectional study (implementation)	Rationale, Scope	Primary care level
			Population: People living with HIV/AIDS		
			Health workers: Health surveillance assistants (HSAs)		
Hoke 2012	Madagascar	Community-based provision of injectable contraceptives in Madagascar: ‘task shifting’ to expand access to injectable contraceptives	Design: Cross-sectional study (implementation)	Rationale, Scope	Community care level
			Population: Women of reproductive age requiring contraceptives		
			Health workers: Community-based distribution (CBD) agents/community health workers		
Born 2012	Zambia	Evaluation of a task-shifting strategy involving peer educators in HIV care and treatment clinics in Lusaka, Zambia	Design: Cross-sectional mixed-methods study	Rationale, Scope	Community care level
			Population: People living with HIV/AIDS		
			Health workers: Peer educators (PEs).		
Dambisya 2012	Uganda	Policy and programmatic implications of task shifting in Uganda: a case study	Design: Cross-sectional, qualitative descriptive study	Rationale, Scope	Secondary, primary, and community care level
			Population: General population, PLWHA, etc.		
			Health workers: Clinical officers, nurses, midwives, and CHWs		
Mafigiri 2012	Uganda	Task shifting for tuberculosis control: A qualitative study of community-based directly observed therapy in urban Uganda	Design Cross-sectional mixed-methods study	Rationale, Scope	Community care level
			Population: People living receiving TB care		
			Health workers: Laypersons		
Kiweewa 2013	Uganda	Noninferiority of a task-shifting HIV care and treatment model using peer counselors and nurses among Ugandan women initiated on ART: evidence from a randomized trial	Design: Prospective randomized intervention trial study	Rationale, Scope	Secondary care level
			Population: Women on ART		
			Health workers: peer counselors and nurses		
Boullé 2013	Cameroon	Task shifting HIV care in rural district hospitals in Cameroon	Design cohort study	Rationale, Scope	Secondary care level
			Population: People living with HIV/AIDS		
			Health workers: Nurses		
Ledikwe 2013	Botswana	Evaluation of a well-established task-shifting initiative: The lay counselor cadre in Botswana	Design: Cross-sectional multi-method study	Rationale, Scope	Primary and community care level
			Population: People living with HIV/AIDS		
			Health workers: Lay Counselors		
Galukande 2013	Uganda	Use of surgical task shifting to scale up essential surgical services: a feasibility analysis at facility level in Uganda	Design: Cross-sectional qualitative study	Rationale, Scope	Secondary care level
			Population General population		
			Health workers: surgical specialists		
Baine 2014	Uganda	A scoping study on task shifting: the case of Uganda	Design: Cross-sectional qualitative study	Rationale, Scope	Secondary, primary, and community care levels
			Population: General population		
			Health workers: Clinical officers, nurses		
Asfaw 2014	Ethiopia	Patient satisfaction with task shifting of antiretroviral services in Ethiopia: implications for universal health coverage	Design: Cross-sectional study	Scope	Secondary and primary care level
			Population: People living with HIV/AIDS		
			Health workers: Health officers and nurses		
Paul 2014	Uganda	Barriers and facilitators in the provision of post-abortion care at district level in central Uganda—a qualitative study focusing on task sharing between physicians and midwives	Design: Cross-sectional qualitative study	Rationale, Scope	Secondary care level
			Population: Women of reproductive health		
			Health workers: Midwives		
O’Malley 2014	Namibia	Nurse task shifting for antiretroviral treatment services in Namibia: implementation research to move evidence into action	Design: Cross-sectional study (implementation)	Rationale, Scope	Secondary and primary care level
			Population: People living with HIV/AIDS		
			Health workers: Nurses		
Andriamanjato 2014	Madagascar, Malawi, and Rwanda	Task shifting in primary eye care: how sensitive and specific are common signs and symptoms to predict conditions requiring referral to specialist eye personnel?	Design: Cross-sectional study (implementation)	Rationale, Scope	Secondary care level
			Population: General population		
			Health workers: General primary healthcare (PHC) workers—ophthalmic clinical officers		
Eliah 2014	Kenya, Malawi, and Tanzania	Task shifting for cataract surgery in eastern Africa: productivity and attrition of non-physician cataract surgeons in Kenya, Malawi, and Tanzania	Design: Cross-sectional study (implementation)	Rationale, Scope	Secondary care level
			Population: General population		
			Health workers: Non-physician cataract surgeons		
Wiedenmayer 2015	Tanzania	The reality of task shifting in medicines management- a case study from Tanzania	Design: Cross-sectional study	Rationale, Scope	Secondary and primary care level
			Population: General population		
			Health workers: Nurses and medical attendants		
Mwangala 2015	Zambia	Task-shifting and quality of HIV testing services: experiences from a National Reference Hospital in Zambia	Design: Cross-sectional qualitative study	Rationale, Scope	Secondary care level
			Population: General population.		
			Health workers: Lay counselors, nurses, and laboratory personnel		
Suzan-Monti 2015	Cameroon	Benefits of task-shifting HIV care to nurses in terms of health-related quality of life in patients initiating antiretroviral therapy in rural district hospitals in Cameroon	Design: Cross-sectional study (implementation)	Rationale, Scope	Secondary care level
			Population: People living with HIV/AIDS		
			Health workers: Nurses		
Chamberlain 2015	Uganda	Mortality related to acute illness and injury in rural Uganda: task shifting to improve outcomes	Design: Cross-sectional study (intervention)	Rationale, Scope	Secondary care level
			Population: General population		
			Health workers: Nurses		
Charyeva 2015	Nigeria	Task shifting provision of contraceptive implants to community health extension workers: results of operations research in northern Nigeria	Design: Cross-sectional study (intervention)	Rationale, Scope	Community care level
			Population: Women of reproductive age		
			Health workers: Community health extension workers		
Agyapon 2015	Ghana	Task shifting Ghana’s community mental health workers’ experiences and perceptions of their roles and scope of practice	Design: Cross-sectional study	Rationale, Scope	Community care level
			Population: General population		
			Health workers: Community mental health workers—community psychiatric nurses (CPNs), clinical psychiatric officers (CPOs), and community mental health officers (CMHOs)		
Akeju 2016	Nigeria	Human resource constraints and the prospect of task-sharing among community health workers for the detection of early signs of pre-eclampsia in Ogun State, Nigeria	Design: Cross-sectional qualitative study	Rationale, Scope	Primary care level
			Population: Women of reproductive age		
			Health workers: Community health extension workers (CHEW)		
Wright 2016	Malawi	Building capacity for community mental health care in rural Malawi: Findings from a district-wide task-sharing intervention with village-based health workers	Design: Cross-sectional study (intervention)	Rationale, Scope	Primary and community care level
			Population: General population		
			Health workers: Village-based health workers		
Gueye 2016	Senegal	Mentoring, task sharing, and community outreach through the TutoratPlus approach: increasing use of long-acting reversible contraceptives in Senegal	Design: Cross-sectional study (intervention)	Rationale, Scope	Primary and community care level
			Population: Women of reproductive age		
			Health workers: Nurses, non-clinical family planning counselors, and community health workers		
Dos Santos 2016	Mozambique	Overview of the mental health system in Mozambique: addressing the treatment gap with a task-shifting strategy in primary care	Design: Cross-sectional study (intervention)	Rationale, Scope	Primary and community care level
			Population: General population		
			Health workers: Psychiatric technicians		
Agyapong 2016a	Ghana	Improving Ghana’s mental healthcare through task-shifting psychiatrists’ and health policy directors’ perceptions about government’s commitment and the role of community mental health workers	Design: Cross-sectional study (mixed methods)	Rationale, Scope	Primary and community care level
			Population: General population		
			Health workers: Community mental health workers—community mental health officers (CMHOs), clinical psychiatric officers (CPOs), and clinical psychiatric nurses (CPNs)		
Some 2016	Kenya	Task shifting the management of non-communicable diseases to nurses in Kibera, Kenya: does it work?	Design: Cross-sectional study (descriptive retrospective)	Rationale, Scope	Primary care level
			Population: General population		
			Health workers: Nurses		
Agyapong 2016b	Ghana	Task shifting perception of stake holders about adequacy of training and supervision for community mental health workers in Ghana	Design: Cross-sectional study (mixed methods)	Rationale, Scope	Primary and community care level
			Population: General population		
			Health workers: Community mental health workers—community mental health officers (CMHOs), clinical psychiatric officers (CPOs) and, clinical psychiatric nurses (CPNs)		
Landes 2017	Malawi	Task shifting of triage to peer expert informal care providers at a tertiary referral HIV clinic in Malawi: a cross-sectional operational evaluation	Design Cross-sectional study (implementation)	Rationale, Scope	Tertiary care level
			Population: People living with HIV/AIDS		
			Health workers: Lay health cadre of expert patients (EPs)		
Okyere 2017	Ghana	Is task-shifting a solution to the health workers’ shortage in Northern Ghana?	Design: Cross-sectional study (qualitative)	Rationale	All levels
			Population: General population		
			Health workers: Medical assistants (MA), midwives, general registered nurses (GRN), enrolled nurses (EN), community health officers (CHO), disease control officers (DCO), psychiatric nurses (PN), optometrist, and health nurse aides		
Tilahun 2017	Ethiopia	Improving contraceptive access, use, and method mix by task sharing Implanon insertion to frontline health workers: the experience of the integrated family health program in Ethiopia	Design: Cross-sectional study (intervention)	Rationale, Scope	Primary and community care level
			Population: Women of reproductive age		
			Health workers: Health extension workers		
Bolkan 2017	Sierra Leone	Safety, productivity and predicted contribution of a surgical task-sharing programme in Sierra Leone	Design: Prospective observational study	Rationale, Scope	Secondary care level
			Population: General population		
			Health workers: Associate clinicians/community health officers (CHOs) and junior doctors		
Farley 2017	South Africa	Evaluation of a nurse practitioner–physician task-sharing model for multidrug-resistant tuberculosis in South Africa	Design: Prospective cohort study	Rationale, Scope	Secondary care level
			Population: People with multidrug-resistant tuberculosis		
			Health workers: Clinical nurse practitioner (CNP) and a medical officer (MO)		
Lulebo 2017	Democratic Republic of the Congo	Task shifting in the management of hypertension in Kinshasa, Democratic Republic of Congo: a cross-sectional study	Design: Cross-sectional study	Rationale, Scope	Primary care level
			Population: People with hypertension		
			Health workers: Nurses		
Gyamfi 2017	Ghana	Training nurses in task-shifting strategies for the management and control of hypertension in Ghana: a mixed-methods study	Design: Mixed-methods study	Rationale, Scope,	Secondary, and primary care level
			Population: People with hypertension		
			Health workers: Community health nurses (CHNs) and enrolled nurses (ENs)		
Dlamini-Simelane 2017	Swaziland	Task shifting or shifting care practices? The impact of task shifting on patients’ experiences and health care arrangements in Swaziland	Design: Cross-sectional (qualitative—ethnographic) study	Rationale, Scope	Primary and community care level
			Population: People living with HIV/AIDS		
			Health workers: Lay counsellors and nurses		
Kaindjee-Tjituka 2017	Namibia	Task-shifting point-of-care CD4+ testing to lay health workers in HIV care and treatment services in Namibia	Design: Cross-sectional study	Rationale, Scope	Primary and community care level
			Population: People living with HIV/AIDS		
			Health workers: Lay health workers and nurses		
Naburi 2017	Tanzania	The potential of task-shifting in scaling up services for prevention of mother-to-child transmission of HIV: a time and motion study in Dar es Salaam, Tanzania	Design: Cross-sectional study (intervention)	Rationale, Scope	Primary and community care level
			Population: People living with HIV/AIDS		
			Health workers: Community health workers (CHWs) and nurses		
Naikoba 2017	Uganda	Improved HIV and TB knowledge and competence among mid-level providers in a cluster-randomized trial of one-on-one mentorship for task shifting	Design: Cluster-randomized trial	Rationale, Scope	Secondary and primary care level
			Population: People living with HIV/AIDS		
			Health workers: Mid-level providers (MLPs)—clinical officers, registered nurses, and registered midwives		
Marotta 2018	Mozambique	Pathways of care for HIV-infected children in Beira, Mozambique: pre–post intervention study to assess impact of task shifting	Design: Cross-sectional study (intervention)	Rationale, Scope	Primary and community care level
			Population: People living with HIV/AIDS		
			Health workers: Maternal and child nurses		
Davis 2018	Malawi	Task shifting levonorgestrel implant insertion to community midwife assistants in Malawi: results from a non-inferiority evaluation	Design: Cross-sectional study (intervention)	Rationale, Scope	Primary and community care level
			Population: Women of reproductive age requiring contraceptives		
			Health workers: Community midwife assistants (CMAs)		
Awolude 2018	Nigeria	Screen and triage by community extension workers to facilitate screen and treat: task-sharing strategy to achieve universal coverage for cervical cancer screening in Nigeria	Design: Cross-sectional study (intervention)	Rationale, Scope	Primary care level
			Population: Women of reproductive age		
			Health workers: CHEWs and CHOs		
Sayed 2018	Kenya	Task Sharing and shifting to provide pathology diagnostic services: The Kenya fine-needle aspiration biopsy cytology and bone marrow aspiration and trephine biopsy training program	Design: Cross-sectional study (intervention)	Rationale, Scope	Tertiary care level
			Population: General population		
			Health workers: Pathologists, medical officers (MO), clinical officers (CO), and technologists		
Millogo 2019	Burkina Faso	Task sharing for family planning services, Burkina Faso	Design: Cross-sectional study (intervention)	Rationale, Scope	Primary and community care level
			Population: Women of reproductive age requiring contraceptives		
			Health workers: Community health workers (CHWs), auxiliary nurses, and auxiliary midwives		
Tariku 2019	Ethiopia	Surgical task shifting helps reduce neonatal mortality in Ethiopia: A retrospective cohort study	Design: Retrospective cohort study	Rationale, Scope	Secondary care level
			Population: Women of reproductive age requiring contraceptives		
			Health workers: Non-physician surgeons (NPS)		
Wall 2020	Kenya	What about lay counselors’ experiences of task-shifting mental health interventions? Example from a family-based intervention in Kenya	Design: Cross-sectional study (mixed methods)	Scope	Community care level
			Population: General population		
			Health workers: Lay counselors		
Gbagbo 2020	Ghana	Increasing access to intrauterine contraceptive device uptake in Ghana: stakeholders’ views on task sharing service delivery with community health nurses	Design: Cross-sectional qualitative study	Rationale, Scope	Community care level
			Population: Women of reproductive age requiring contraceptives		
			Health workers: Community health nurses		
Peresu 2020	Eswatini	Task-shifting directly observed treatment and multidrug-resistant tuberculosis injection administration to lay health workers: stakeholder perceptions in rural Eswatini	Design: Mixed-methods study—cross-sectional survey and qualitative study	Rationale, Scope	Community care level
			Population: People with tuberculosis		
			Health workers: Lay health workers (LHWs)/community treatment supporter (CTS)		
Lund 2020	South Africa	Task-sharing of psychological treatment for antenatal depression in Khayelitsha, South Africa: Effects on antenatal and postnatal outcomes in an individual randomized controlled trial	Design: Randomized controlled trial	Rationale, Scope	Primary and community care level
			Population: Women of reproductive age		
			Health workers: Non-specialist community health workers		
Sevene 2021	Mozambique	Feasibility of task-sharing with community health workers for the identification, emergency management, and referral of women with pre-eclampsia, in Mozambique	Design: Mixed-methods study	Rationale, Scope	Primary and community care level
			Population: Pregnant women		
			Health workers: Community health workers		
Jacobs 2021	South Africa	Task sharing or task dumping: counsellors experiences of delivering a psychosocial intervention for mental health problems in South Africa	Design: Cross-sectional qualitative study	Scope	Primary care level
			Population: General population		
			Health workers: Non-specialist—facility-based counsellors (FBCs), specific cadre of community health workers trained to deliver health promotion and HIV adherence counselling services		
Yator 2021	Kenya	Task-sharing and piloting WHO group interpersonal psychotherapy (IPT-G) for adolescent mothers living with HIV in Nairobi primary health care centers: a process paper	Design: Cross-sectional study (intervention)	Scope	Primary care level
			Population: Postpartum adolescent (PPA) mothers living with HIV		
			Health workers: Community health workers (CHWs)		

**Table 5 healthcare-11-01200-t005:** Main theme and sub-themes.

Main Theme	Sub-Themes
Rationale for task shifting and task sharing	Health worker shortages
	Optimally utilize existing health workers
	Expand access to health services
Scope of shifted and shared tasks	HIV/AIDS care
	Hypertension management
	Diabetes management
	Mental health
	Maternal and child healthcare
	Sexual and reproductive health services
	Eye care
	Tuberculosis care
	Surgical care and procedures
	Medicines’ management
	Emergency care

**Table 6 healthcare-11-01200-t006:** Summary of key findings on rationale and scope of task shifting and task sharing in reviewed studies.

First Author (Year)	Country	Health Services Context	Key Findings
Bemelmans 2010	Malawi	Access to HIV⁄AIDS care	Rationale: Staff shortages in the levels of care and by location. Scope: HIV testing and counselling from nurses to health surveillance assistant (HSA)/lay counsellor, antiretroviral therapy (ART) initiations from medical doctors to non-physician clinicians—clinical officer, medical assistant, and nurse
Kosgei 2010	Kenya	Access to HIV⁄AIDS care	Rationale: Scarcity of healthcare providers and the need to improve patient outcomes without increasing clinic human resources. Scope: ART care
Labhardt 2010	Cameroon	Access to care for hypertension and type 2 diabetes care	Rationale: Majority of the rural population does not have access to adequate hypertension and diabetes care. Scope: Hypertension and diabetes care
Selke 2010	Kenya	Access to HIV⁄AIDS care	Scope: Delivery of medications and provision of follow-up care to patients on ART in the community with support of an electronic decision tool, and 3 monthly visits to facilities compared to the usual monthly visit.
De Wet 2011	South Africa	Access to HIV⁄AIDS care	Rationale: Shortage of health workers (physicians and nurses). Scope: Task shifting from nurses to community health workers for HIV treatment and care. HIV counselling, drug readiness training, distribution of nutritional supplements, and capturing and updating electronic information.
Chibanda 2011	Zimbabwe	Access to mental health intervention/services	Scope: Depression and other common mental disorders (CMD)—screening and monitoring CMD and in delivering the intervention
Gessessew 2011	Ethiopia	Access to comprehensive obstetric care	Rationale: Shortage of physicians in rural areas. Scope: Comprehensive emergency obstetric care (CEmOC)
Jennings 2011	Benin	Access to maternal and newborn health services	Rationale: Need to expand the role of lay nurse aides. Scope: Counselling in maternal and newborn care.
Umar 2011	Nigeria	Access to HIV⁄AIDS care	Rationale: Long waiting time of HIV/AIDS patients in the clinic due to the high workload of available doctors. Scope: Consultation for HIV patients presenting for routine refill and follow-up visits
Tweya 2012	Malawi	Access to HIV⁄AIDS care	Rationale: Shortage of clinicians and nurses. Scope Provision of antiretroviral therapy (ART) to stable patients.
Hoke 2012	Madagascar	Access to injectable contraceptives	Rationale Lack of access to health facilities. Scope: Injection (re-injection) and counselling of patients.
Born 2012	Zambia	Access to HIV⁄AIDS care	Rationale: Rapid expansion of antiretroviral therapy (ART) using existing health workers. Scope: Provision of counselling, education talks, and adherence support to patients in HIV care.
Dambisya 2012	Uganda	Access to HIV⁄AIDS care, maternal and child health, general healthcare, etc.	Rationale: Severe health worker shortage and a high demand for healthcare services. Scope: Community health workers (CHW) and PLWHA in care and support of AIDS patients, ophthalmic clinical officers conduct cataract surgery, psychiatric clinical officers cover the same scope as the psychiatrists, but are more community-oriented than the psychiatrists, who tend to be mainly hospital-based. Nurses set IV lines in upcountry due to lack of physicians, midwives conduct manual vacuum extraction, manual removal of the placenta, and manual vacuum aspiration due to shortage of doctors. CHWs and community members involved in delivery of expanded program on immunization (EPI) services, etc.
Mafigiri 2012	Uganda	Access to TB care	Rationale: To address barriers to successful DOTS in rural areas. Scope: Directly observed treatment short course (DOTS)
Kiweewa 2013	Uganda	Access to HIV⁄AIDS care	Rationale Shortage of physicians. Scope: ART follow-up care to postpartum women
Boullé 2013	Cameroon	Access to HIV⁄AIDS care	Rationale Shortage of physicians
Ledikwe 2013	Botswana	Access to HIV⁄AIDS care	Rationale: Shortage of health workers. Scope: HIV tests and related counselling at public health facilities
Galukande 2013	Uganda	Access to surgical services	Rationale: Shortage and maldistribution of surgical specialists. Scope: Emergency and essential surgical care
Baine 2014	Uganda	Access to health services	Rationale: Shortage of health workers in Uganda. Scope: Surgical care, sexual and reproductive health, HIV/AIDS, tuberculosis DOTS therapy.
Asfaw 2014	Ethiopia	Access to HIV⁄AIDS care	Scope: Antiretroviral therapy
Paul 2014	Uganda	Access to post-abortion care	Rationale Absence of physicians. Scope: Post-abortion care
O’Malley 2014	Namibia	Access to HIV⁄AIDS care	Rationale: Shortage of physicians. Scope: Antiretroviral treatment services
Andriamanjato 2014	Madagascar, Malawi, and Rwanda	Access to eye care	Rationale: Shortage of health workers trained in eye care. Scope: Primary eye care
Eliah 2014	Kenya, Malawi, and Tanzania	Access to eye care	Rationale: Shortage and maldistribution of ophthalmologists. Scope: Cataract surgery
Wiedenmayer 2015	Tanzania	Access to medicines	Rationale: Severe shortage of pharmaceutical staff. Scope: Pharmaceutical management
Mwangala 2015	Zambia	Access to HIV testing	Rationale: Shortage of human resources. Scope: HIV testing services
Suzan-Monti 2015	Cameroon	Access to HIV⁄AIDS care	Rationale: Shortage of physicians in rural areas. Scope: Initiating antiretroviral therapy (ART)
Chamberlain 2015	Uganda	Access to emergency care services	Rationale: Critical shortages of acute care and healthcare workers in resource-limited settings. Scope: Emergency care services in a rural setting
Charyeva 2015	Nigeria	Access to contraceptive implants	Rationale: Severe shortage of human resources. Scope: Provision of contraceptive implants
Agyapon 2015	Ghana	Access to mental health services	Rationale: Inadequate numbers of psychiatrists. Scope: Mental healthcare
Akeju 2016	Nigeria	Access to maternal health services	Rationale: Non-availability of health personnel at the primary healthcare level. Scope: Detection of early signs of pre-eclampsia
Wright 2016	Malawi	Access to mental health services	Rationale: Improve access to primary mental healthcare. Scope: Primary mental healthcare
Gueye 2016	Senegal	Access to family planning services	Rationale: To improve access to family planning in rural areas and improve contraceptive prevalence. Scope: Provision of family planning services—long-acting reversible contraceptives (LARC), specifically implants and the intrauterine device.
Dos Santos 2016	Mozambique	Access to mental health services	Rationale: To expand access to primary mental healthcare due to low numbers of psychiatrists and psychologists Scope: Delivery of psychiatric care
Agyapong 2016a	Ghana	Access to mental health services	Rationale: Expand mental healthcare delivery due to shortage of psychiatrists. Scope: CPNs and CMHOs are primarily responsible for case detection in the community, referral of patients to CPOs and psychiatrists. CPOs are responsible for diagnosing and treating a range of common psychiatric conditions
Some 2016	Kenya	Access to non-communicable diseases (NCDs) care	Rationale: Shortage of health workers at the primary level of care and need to increase access to NCD care in primary healthcare settings. Scope: Management of NCDs (hypertension, diabetes mellitus type 2, epilepsy, asthma, and sickle cell)
Agyapong 2016b	Ghana	Access to mental health services	Rationale: Shortage of psychiatrists. Scope: Case detection and referral, and diagnosis and treatment of common psychiatric conditions.
Landes 2017	Malawi	Access to HIV⁄AIDS care	Rationale: Shortage of health workers Scope: Triaging of HIV/AIDS patients
Okyere 2017	Ghana		Rationale: Insufficient health workers.
Tilahun 2017	Ethiopia	Access to family planning services	Rationale: To improve access to family planning in rural areas and improve contraceptive prevalence. Scope: Provision of long-acting contraceptive (Implanon) family planning services.
Bolkan 2017	Sierra Leone	Access to emergency surgical care	Rationale: Shortage of surgical providers. Scope: Surgical and obstetric emergencies
Farley 2017	South Africa	Access to TB treatment services	Rationale: Shortages of medical officers to implement decentralization of MDR-TB treatment service to outpatient settings/lower-level health facilities. Scope: Multidrug-resistant tuberculosis (MDR-TB) treatment
Lulebo 2017	Democratic Republic of the Congo	Access to hypertension management services	Rationale: Shortage of health workers. Scope: Hypertension management
Gyamfi 2017	Ghana	Access to hypertension management services	Rationale: To expand service delivery to lower levels. Scope: Hypertension management and control
Dlamini-Simelane 2017	Swaziland	Access to HIV⁄AIDS care	Rationale: Decentralized ART provision to improve access. Scope: HIV counselling by lay counsellors (predominantly PLHIV), initiation of patients on ART by nurses
Kaindjee-Tjituka 2017	Namibia	Access to CD4+ testing	Rationale: Roll-out and scale-up of POC CD4+ testing in HCT settings in public health facilities. Scope: Point-of-care (POC) CD4+ testing
Naburi 2017	Tanzania	Access to HIV⁄AIDS care—PMTCT services	Rationale: Reduce nurses’ workload and health system costs. Scope: Prevention of mother-to-child transmission of HIV (PMTCT) service delivery
Naikoba 2017	Uganda	Access to HIV/AIDS and TB services	Rationale: Health worker shortages. Scope: HIV care and treatment
Marotta 2018	Mozambique	Access to HIV/AIDS services for children < 5 years	Rationale: To improve ART initiation and retention of HIV-infected children. Scope: Care for HIV-positive children < 5 years old
Davis 2018	Malawi	Access to contraceptive implants	Rationale: Shortage of nurses/midwives. Scope: Long-acting reversible contraception (LARC) insertion (levonorgestrel (LNG) contraceptive implants)
Awolude 2018	Nigeria	Access to cervical cancer screenings	Rationale: Shortage of doctors and nurses in rural areas. Scope: Screen for cervical cancer using visual inspection with acetic acid
Sayed 2018	Kenya	Access to cancer screenings	Rationale: Scarcity of pathologists in Kenya. Scope: Fine-needle aspiration biopsy cytology and bone marrow aspiration and trephine biopsy
Millogo 2019	Burkina Faso	Access to contraceptives	Rationale Shortage of qualified health staff. Scope: Community health workers (CHWs) to offer oral and injectable contraceptives to new users, and auxiliary nurses and auxiliary midwives to provide implants and intrauterine devices.
Tariku 2019	Ethiopia	Access to contraceptives	Rationale: Shortage of physicians, improve access to surgical services and reduce neonatal mortality. Scope: Caesarean section
Wall 2020	Kenya	Access to mental health services	Rationale: Improve access at the community level. Scope: Community-based family therapy mental health interventions
Gbagbo 2020	Ghana	Access to intrauterine contraceptive device	Rationale: Addressing persistent human resources shortages. Scope: Intrauterine contraceptive device
Peresu 2020	Eswatini	Access to multidrug-resistant tuberculosis (MDR-TB) treatment in rural areas	Rationale: Shortage of human resources for health (HRH) and limited access to multidrug-resistant tuberculosis (MDR-TB) treatment in rural areas. Scope: Directly observed treatment (DOT) supervision and administration of intramuscular MDR-TB injections
Lund 2020	South Africa	Access to mental health services	Rationale: Dearth of mental health professionals. Scope: Psychological treatment for perinatal depression
Sevene 2021	Mozambique	Access to hypertension services	Rationale: Delays in reaching health facilities and insufficient healthcare professionals. Scope: Initial screening and initiation of obstetric emergency care for pre-eclampsia/eclampsia
Jacobs 2021	South Africa	Access to mental health services	Scope: Mental health counselling
Yator 2021	Kenya	Access to HIV/AIDS care	Scope: Interpersonal psychotherapy (IPT-G) for adolescent mothers living with HIV

## Data Availability

Data sharing is not applicable to this article.

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
