# Peer review of "Task Shifting and Task Sharing Implementation in Africa: A Scoping Review on Rationale and Scope"

_healthcare, 2023, doi:10.3390/healthcare11081200_

Round 1
Reviewer 1 Report
This manuscript explores and synthesizes the evidence about scope of task shifting and task sharing in Africa. This is a study with a theme and methods that meet the scope of a scientific journal. Although the study is very interesting and of good quality, this reviewer considers that more information is needed for its publication.
In the methods section, it is necessary to review the presented exclusion criteria 1 and 2. Exclusion criteria do not represent the opposite of inclusion criteria. An important methodological limitation of the study is the fact that only articles in English were considered, when there are African countries whose official languages are French, Portuguese or other languages. Even though knowledge production is massively published in English, many interesting publications may have been overlooked.
In the results section, a first aspect to be reviewed is the writing structure. The recurrent enunciations of different studies make the text quite tiring (Ex. Three studies in Ghana reported [...] A study in Malawi reported [...] A study in ...). A more direct and fluid text is recommended. In addition, much repeated information is contained in the illustrations and text. Wouldn't it be a case of simplifying the text to make room for other evidence? For example, what strategies enabled task shifting and task sharing in Africa? Was any legislation necessary? Are there already measurable results from implementing these approaches?
The discussion section overlaps the results section. A lot of repeated information can give rise to a deeper analysis on the subject, especially with regard to the limitations related to the implementation of alternation and task sharing strategies in Africa.
Author Response
The initial manuscript was peer-reviewed and we thank the reviewers for their feedback. We have provided responses to their feedback below and made relevant changes in the manuscript, and they are tracked.
Many thanks for your kind consideration.
Please see attached

Reviewer 2 Report
Your manuscript is clear and the tables you used are informative. To make it easy for the reader, I suggest you re-position the tables as follows: Table 1 be moved to Line 77; Table 2 to Line 83; Table 3 to Line 136; Table 4 to Line 137 and Table 5 to Line 143. Your themes and sub-themes are clear, however, I suggest that you display them in a table to give the reader an overview before going through them in the text. You have clearly defined the main themes to make the reader understand what they are about, however, these definitions are missing for the sub-themes. Beside capacity building for beneficiary cadres (Lines 357 and 408), incentives should also be considered.
Author Response

(The authors gave the same response as above.)
